# Low-Dose rIL-15 Protects from Nephrotoxic Serum Nephritis via CD8^+^ T Cells

**DOI:** 10.3390/cells11223656

**Published:** 2022-11-18

**Authors:** Agnes A. Mooslechner, Max Schuller, Katharina Artinger, Alexander H. Kirsch, Corinna Schabhüttl, Philipp Eller, Alexander R. Rosenkranz, Kathrin Eller

**Affiliations:** 1Division of Nephrology, Department of Internal Medicine, Medical University of Graz, 8036 Graz, Austria; 2Intensive Care Unit, Department of Internal Medicine, Medical University of Graz, 8036 Graz, Austria

**Keywords:** glomerulonephritis, interleukin-15, renal tubular epithelial cell

## Abstract

Rapid progressive glomerulonephritis (GN) often leads to end-stage kidney disease, driving the need for renal replacement therapy and posing a global health burden. Low-dose cytokine-based immunotherapies provide a new strategy to treat GN. IL-15 is a strong candidate for the therapy of immune-mediated kidney disease since it has proven to be tubular-protective before. Therefore, we set out to test the potential of low-dose rIL-15 treatment in a mouse model of nephrotoxic serum nephritis (NTS), mimicking immune complex-driven GN in humans. A single low-dose treatment with rIL-15 ameliorated NTS, reflected by reduced albuminuria, less tissue scarring, fewer myeloid cells in the kidney, and improved tubular epithelial cell survival. In addition, CD8^+^ T cells, a primary target of IL-15, showed altered gene expression and function corresponding with less cytotoxicity mediated by rIL-15. With the use of transgenic knock-out mice, antibody depletion, and adoptive cell transfer studies, we here show that the beneficial effects of rIL-15 treatment in NTS depended on CD8^+^ T cells, suggesting a pivotal role for them in the underlying mechanism. Our findings add to existing evidence of the association of IL-15 with kidney health and imply a potential for low-dose rIL-15 immunotherapies in GN.

## 1. Introduction

End-stage kidney disease is a global health problem with a steady increase in patients requiring kidney replacement therapy. In the western world, up to 20% of cases are attributed to glomerulonephritis (GN), which is often associated with autoimmunity and for which new therapeutic options are required [1,2]. GN is characterized by glomerular hypercellularity, causing immune-mediated damage and fibrosis [3]. In addition, acute tubular injury and interstitial nephritis often accompany GN and are associated with worse outcomes [4,5].

Cytokines are potent mediators of immune signaling and pose great potential for immunotherapies. Previously cytokines like IL-2 have been used in high doses to stimulate inflammatory immune responses against pathogens and malignancies [6]. However, treatments effective for autoimmune diseases later evolved to low-dose protocols, aiming at regulatory CD4^+^ T cells and their high-affinity receptors [7]. Amongst others, low-dose IL-2 was successfully used in systemic lupus erythematosus [8,9] and type 1 diabetes mellitus [10,11]. 

IL-2 and IL-15 are members of the same cytokine family, the α-helix bundle family, and are closely related. Both cytokines have unique Rα-chains but share the same Rβ- and Rγ-chains, explaining some overlapping functions. IL-15 is a key factor for CD8^+^ T cell and NK cell proliferation and cytotoxicity [12]. These characteristics are tested in clinical trials as combination immunotherapy against cancers [13]. However, like IL-2, IL-15 is a pleiotropic cytokine, and its receptors are found on various cell types. In the kidney, IL-15 mRNA was found to be expressed in mice and human tubular epithelial cells (TECs) [14,15]. Additionally, all three receptor chains were shown to be present on renal proximal TECs, allowing the presentation of IL-15 in *trans* to responding cell types [16]. In vitro cell line studies showed increased TEC survival mediated by rIL-15 [17]. The same study reported that *IL-15^−/−^* mice were more susceptible to autoimmune diseases and nephrotoxic serum nephritis. Additionally, IL-15 has been found to counteract renal fibrosis [18]. In line with these preclinical data, transcriptomic datasets of kidney biopsies from patients with different nephropathies (including IgA nephropathy, segmental and membranous glomerulonephritis) revealed lower IL-15 expression than in healthy controls [19] and the same results were shown on protein level. This study highlights the importance of homeostatic levels of IL-15 in kidney health, making it a potential candidate for immunotherapies in GN.

Nephrotoxic serum nephritis (NTS) is a mouse model of immune complex-mediated GN, mimicking forms of rapid-progressive GN in humans. Like in human biopsies, also NTS mice showed reduced IL-15 expression in kidney tissue compared to controls [17]. The disease is mediated among other players by CD4^+^ T cells [20,21,22] and regulated by CD4^+^ regulatory T cells [23,24,25,26]. CD8^+^ T cells were shown to migrate independently of CD4^+^ T cells into renal tissues [22]. Even though they have long been studied in NTS, their distinct role in the pathophysiology is still unclear [27]. Studies in mice report contradicting results, showing both benefit and harm of CD8 knock-out and depletion.

While high doses of IL-15 stimulate inflammatory CD8^+^ T cell immune responses [12,28,29], there is a lack of knowledge regarding low-dose IL-15 treatment. Therefore, given the success of low-dose IL-2 therapies in autoimmunity, we set out to test low-dose rIL-15 as treatment in a mouse model of NTS, with a specific focus on TEC health and CD8^+^ T cell involvement.

## 2. Materials and Methods

### 2.1. Animal Studies

C57BL6/J mice were purchased from Charles River, Germany. B6.129S2-Cd8atm1Mak/J2 (*CD8α^−/−^*) were purchased from The Jackson Laboratory, Bar Harbor, ME, USA. All mice were bred and maintained in a pathogen-free environment at the Medical University of Graz, Austria. All experiments were approved by the Federal Ministry of Education, Science, and Research of the Republic of Austria (2020-0.078.387). All experiments were performed on mice ranging from 8 to 14 weeks, using littermate-, age- and sex-matched controls.

### 2.2. In Vivo Interventions

NTS model: Mice were injected with rabbit IgG antibody in complete Freud’s Adjuvant s.c. in the footpad. Depending on the experiment setup, three or five days later, NTS was induced by i.v. injection with nephrotoxic rabbit anti-mouse serum [30]. Mice were kept for 7 days. 

Injections: Low-dose rIL-15 was administered via a single i.p. injection in a 0.1 µg/g bodyweight concentration [31,32,33]. Anti-CD8α (Clone YTS 169.4, BioXCell, Lebanon, NH, USA) depletion antibody and respective isotype control (Clone LTF-2, BioXCell, Lebanon, NH, USA) were injected i.p. in a concentration of 200 µg/20 g bodyweight twice per week.

### 2.3. Assessment of Renal Injury

Blood Urea Nitrogen (BUN) was measured via colorimetric assay (Invitrogen, Carlsbad, CA, USA) from plasma according to standard protocol. 

Urinary albumin was assessed by double-sandwich ELISA (Bethyl Laboratories, Montgomery, TX, USA). Creatinine was measured using a picric acid-based method (Sigma-Aldrich, St. Louis, MI, USA) with an enzymatic assay kit (Crystal Chem, Elk Grove Village, IL, USA) or liquid chromatography-tandem mass spectrometry.

Glomerulosclerosis (PAS-Score), crescents, and tubular cast formation were analyzed from formalin-fixed renal tissue embedded in paraffin on 4 µm sections stained with Periodic Acid-Schiff reaction (PAS, Merck KGaA, Darmstadt, Germany). A minimum of 50 equatorial glomerular cross-sections was evaluated and scored regarding the percentage of PAS-positive material per glomerulus. Absolute tubular cast counts are analyzed in 12hpf per sample and identified as acellular PAS-positive cylindrical casts within distal and proximal tubuli. Immune cell infiltration was determined on frozen tissue sections and stained with rat-derived anti-mouse antibodies against CD68 (Bio-Rad Laboratories, Hercules, CA, USA) and Ly6G (Abcam, Cambridge, UK). Cell infiltration was quantified in total cell numbers per six high-power fields (hpf) of renal cortex and medulla.

TUNEL staining was performed on cryosections by terminal transferase dUTP nick-end labeling with Fluorescein with the In Situ Death Detection Kit (Merck, Darmstadt, Germany), according to standard protocol.

Circulating autologous antibody response was detected by serum response to rabbit IgG antibody (Jackson ImmunoResearch Laboratories, West Grove, PA, USA). Circulating mouse anti-rabbit IgG was measured using horseradish peroxidase-conjugated goat anti-mouse IgG (Agilent Technologies, Santa Clara, CA, USA). Histological staining for autologous IgG was achieved by serial dilutions of directly FITC-labeled goat anti-mouse IgG (Jackson ImmunoResearch Laboratories, West Grove, PA, USA) on kidney cryosections.

### 2.4. Multicolor Flow Cytometry Analysis

Lymph node, spleen, and kidney were processed for multicolor flow cytometry analysis. Lymph node and spleen were passed through 70µm nylon mesh to generate single-cell suspensions. Spleen was treated with red blood cell lysis (eBioscience, Thermo Fisher, Waltham, MA, USA). Kidneys were meshed and digested in RPMI media substituted with FBS, β-Mercaptoethanol, HEPES-Buffer, Collagenase D, and DNase I. Tissue was passed through nylon mesh and washed with magnesium- and calcium-free HBSS. The suspension was centrifuged with 37.5% osmo-adjusted percoll, and red blood cells were lysed. Single-cell suspensions were stained in a buffer containing 2mM EDTA and 0.5% BSA. 

All samples were stained with fixable viability dye (eBioscience, Thermo Fisher, Waltham, MA, USA) to exclude dead cells. The following antibodies were used for flow cytometry analysis of CD8^+^ T cells and subpopulations: CD45 (30-F11), CD3 (17A2), CD4 (RM4-5), CD25 (PC61), CD8 (53-6.7), CD44 (IM7), CD122 (TM-b1), Ly49 (14B11); The following antibodies were used to analyze NK, NKT and iNKTS: CD45, CD19 (1D3), CD3, NK1.1 (PK136), CD1d-tet (PBS-57). CD1d-tet antibody was provided by the NIH tetramer core facility at Emory University, Atlanta, GA, USA. The following antibodies were used to analyze T, B, NK, and Lin- cells: CD45, NK1.1, CD19 (6D5), CD3, CD90.2 (30-H12). The following antibodies were used to analyze CD4^+^ T cells and Tregs: CD45, CD3, CD4, CD25, FoxP3 (150D). Intracellular staining was performed with eBioscience FOXP3/Transcription Factor Staining Buffer Set (Invitrogen, Carlsbad, CA, USA). The following antibodies were used to stain for IFNγ expression in CD4 and CD8 T cells: CD45, CD3, CD4 or CD8, IFNγ (XMG1.2). Antibody against CD45 was purchased from BD Bioscience, Franklin Lakes, NJ, USA. Antibodies against CD25 and Foxp3 were purchased from Biolegend, San Diego, CA, USA. All other antibodies were purchased from Thermo Fisher Scientific, Waltham, MA, USA. Cells were fixed in Cytofix (BD Biosciences, Franklin Lakes, NJ, USA) before analysis. Precision count beads (Biolegend, San Diego, CA, USA) were added to determine absolute cell numbers. Data from stained single-cell suspensions were acquired on a CytoFLEX LX (Beckman Coulter, Brea, CA, USA) and analyzed with FlowJo software (FlowJo, LLC and BD Biosciences, Franklin Lakes, NJ, USA).

### 2.5. TEC Isolation and Gene Expression Analysis

TECs were isolated from fresh kidney cortex tissue and meshed on 150-90-45 µm sieves. Tissue was pushed through the top sieve with a plunger and flushed through consecutive sieves with cold PBS. TECs were washed from the 90 µm sieve and frozen in TRIzol Reagent (Sigma-Aldrich, St. Louis, MI, USA).

Total RNA was extracted from kidney or pre-isolated TECs using TRIzol Reagent. cDNA was synthesized using the High-Capacity cDNA Reverse Transcription kit (Applied Biosystems, Waltham, MA, USA). Real-time polymerase chain reaction was performed using TaqMan gene expression assays (Applied Biosystems, Waltham, MA, USA) and SYBR green (Bio-Rad Laboratories, Hercules, CA, USA).

### 2.6. Negative MACS Sort of CD8α^+^ Cells

Single-cell suspensions of lymph nodes were stained with APC-labelled antibodies against CD4 (RM4-5), CD11b (M1/70), CD19 (1D3), TER119 (TER-119), and B220 (RA3-682). Subsequently, cells were incubated with anti-APC microbeads (Miltenyi Biotec, Bergisch Gladbach, Germany). Cell suspensions were put through LD columns, and labeled cells were magnetically bound to the column. Flow-through contained highly enriched CD8^+^ cells and were checked for purity above 95%. For adoptive transfer experiments, 2.5 donor mice per recipient were used. The flow-through contained unstained CD8α^+^ cells. Flow-throughs of all columns were pooled, a purity check was performed, and 2.5 × 106 cells were adoptively transferred into recipient mice.

### 2.7. Cell Culture

To assess the relative gene expression in CD8^+^ T cells, negative MACS sorted cells were cultured (1 × 10^6^ cells per ml, in 2.5 mL, 1 well per sample) in RPMI with 10% FCS, 1%Pen/Strep, and stimulated with Dynabeads^®^ Mouse T-Activator CD3/CD28 (Gibco by Life Technologies, Carlsbad, CA, USA) according to the standard protocol for 4 days. Subsequently, cells were transcribed into cDNA with TaqMan^®^ Gene Expression Cells-to-CT™ Kit (Thermo Fisher Scientific, Waltham, MA, USA) according to standard protocol.

To analyze IFNγ-production by CD4^+^ and CD8^+^ T cells, splenocytes were cultured in RPMI, 10% FCS, 1% Pen/Strep. To analyze CD4^+^ T cells, cells were stimulated with eBioscience Cell Stimulation Cocktail Plus Protein Transport Inhibitor (Invitrogen, Carlsbad, CA, USA) for 5 h, harvested, and processed for flow cytometry. To analyze CD8^+^ T cells, cells were stimulated with Dynabeads^®^ Mouse T-Activator CD3/CD28 (Gibco by LifeTechnologies, USA) according to the standard protocol for 4 days. On day 4, cells were restimulated with eBioscience Cell Stimulation Cocktail Plus Protein Transport Inhibitor (Invitrogen, USA) for 5 h, harvested, and processed for flow cytometry. Cells were cultured as 1 × 10^6^ cells per ml, in 2.5 mL, 2 wells per sample. For intracellular staining, 2 wells per sample were pooled and processed with eBioscience Intracellular Fixation & Permeabilization Buffer Set (Invitrogen, USA) according to the standard protocol. 

### 2.8. Statistical Analysis

Data are shown as mean ± standard error of the mean (SEM). Normal distribution was tested by Kolmogorov–Smirnov test. Comparison between two groups was performed by Student’s *t*-test or nonparametric Mann–Whitney U test. Scores were evaluated with Fisher’s exact test. A comparison of survival curves was tested by Log-rank test. *p*-values of less than 0.05 were considered statistically significant. Statistical analysis was conducted with GraphPad Prism 9.3.1 (GraphPad Software, San Diego, CA, USA). Graphical abstract created with BioRender.com accessed 12 November 2022.

## 3. Results

### 3.1. Low-Dose rIL-15 Treatment Ameliorates NTS

Since high doses of rIL-15 are known to stimulate CD8^+^ cytotoxic cell responses, we set out to study the effect of low-dose rIL-15 as an intervention in NTS. Mice were treated with a low dose of rIL-15, which has been shown to protect tubular epithelial cells in vitro [17], on day 1 of disease. Albuminuria, histological changes, and myeloid cell infiltration were analyzed on day 7 and compared to controls (experimental setup Figure 1A). In the group treated with rIL-15, albuminuria was reduced by 50% (Figure 1B) and histological analysis showed less glomerulosclerosis reflected by lower PAS scores (Figure 1C,D). Interestingly, whereas treatment reduced renal myeloid cell infiltration, represented by fewer Ly6G^+^ cells in glomeruli and the interstitium (Figure 1E) and fewer CD68^+^ cells (Figure 1F), no difference in overall CD3^+^CD4^+^ and CD3^+^CD8^+^ T cell numbers in kidney tissue were found (Figure 1G). 

Of note, BUN levels were not affected by low-dose rIL-15 treatment (Appendix A). Since B cells have been shown to be influenced by IL-15 [34,35], we analyzed the humoral response of control NTS mice versus NTS mice treated with IL-15. We found no difference in circulating mouse anti-rabbit IgG between both groups (Appendix A). Direct fluorescence staining of autologous (Appendix A) and heterologous (data not shown) IgG showed a comparable amount of IgG-deposits on the GBM in both groups. To evaluate the systemic effects of low-dose rIL-15 therapy, we evaluated relevant immune cell populations in the spleen and peripheral blood of mice after 7 days of NTS. Flow cytometry analysis revealed no difference in T cells, B cells, NK cells, and CD3/CD19/NK1.1 lineage negative cells in the spleen (Appendix A). Peripheral blood analysis showed no difference in white blood cell counts and frequencies of granulocytes and monocytes (Appendix A) 

### 3.2. rIL-15 Protects TECs from Cell Death in NTS

IL-15 has been shown to be a survival factor for tubular epithelial cells [17]. Accordingly, low-dose rIL-15 treatment potently reduced tubular injury in our hands, as shown by fewer tubular casts 7 days after NTS induction (Figure 2A). 

Significantly reduced numbers of TUNEL^+^ tubular epithelial cells were found in mice after 7 days of NTS treated with rIL-15 as compared to vehicle (Figure 2B,C). Furthermore, we isolated TECs after 7 days of NTS and investigated their gene expression profile. Analysis showed that in the presence of rIL-15, TECs upregulated the expression of *Il-15rα* and upregulated *Bcl2* (Figure 2D). At the same time, *Mcp1*, *Cxcl1*, *Cxcl5*, and *Nlrp3* expression levels were not altered (Figure 2E).

### 3.3. Administration of rIL-15 in NTS Alters CD8^+^ T Cell Gene Expression Profile and Function

To pinpoint whether enhanced TEC survival is mediated primarily by rIL-15 treatment or rather secondary due to changes in immune cell composition, we evaluated the main target cell populations of IL-15, namely NK and CD8^+^ T cells, first. NK cell numbers were not changed in the spleen due to IL-15 in NTS (Appendix A). Additionally, in kidney tissue, NK, NKT, and iNKT cell numbers were comparable in both groups on day 7 of NTS (Figure 3A–C).

CD8^+^ T cells were found to be increased in the lymph node without reaching significance (Figure 3D). When characterizing them in-depth, we found a significant increase in CD44^+^ memory CD8^+^ T cells (Figure 3E). This increase was also seen in CD122^+^ activated memory CD8^+^ T cells (Figure 3F) and Ly49^+^ activated memory CD8^+^ T cells (Figure 3G). In the kidney, no difference in both CD8^+^ T cell subpopulations, but a trend toward more Ly49^+^ activated memory CD8^+^ T cells (*p* = 0.057, Appendix A) was found. We sorted and cultured CD8^+^ lymphocytes from lymph nodes of mice subjected to 7 days of NTS treated with rIL-15 or vehicle (Appendix A). Analysis of the gene expression profile after 4 days of culture with anti-CD3/anti-CD28 bead stimulation revealed that mice treated with rIL-15 showed a trend for lower *Prf1* expression and significantly reduced expression of *Gzmb* and *Ifnγ* (Figure 3H). Additionally, expression of *Ikzf2* in these cells increased, while there was no change in expression of *Tgfb1* and *Foxp3*. (Figure 3I).

To study IFNγ-production on protein level, we analyzed splenocytes via flow cytometry after 4 days of culture with anti-CD3/anti-CD28 stimulation. We found no difference in CD8^+^ frequencies of CD3^+^ cells after 4 days of culture (Figure 3J). However, the frequencies of IFNγ^+^CD8^+^ cells were reduced in splenocytes isolated from NTS mice treated with rIL-15 compared to controls (Figure 3K,L).

### 3.4. rIL-15 Does Not Increase CD4^+^ Tregs but Induces Higher Rates of IFNγ^+^CD4^+^ T Cells

Since CD4^+^ T cells play a central role in the pathogenesis of NTS and IL-15 affects their proliferation and survival, we next investigated the effect of low-dose rIL-15 on CD4^+^ T cell function and composition. Of note, no differences in mRNA expression of *Tbet*, *Rorc*, and *Foxp3* as markers of Th1, Th17, and Tregs in whole kidney tissue were found (Appendix A). In the lymph node, overall CD4^+^ T cell numbers (Figure 4A) were not changed due to IL-15, and no difference was seen in CD4^+^ Treg frequencies (Figure 4B,C). However, isolated splenocytes of NTS control mice and those treated with rIL-15 stimulated with PMA/Iono showed higher frequencies of IFNγ^+^CD4^+^ T cells (Figure 4D,E). 

### 3.5. rIL-15-Mediated Protection of TECs from Cell Death Is Dependent on CD8^+^ Cells

Since low-dose rIL-15 altered CD8^+^ T cells towards a less cytotoxic phenotype, we aimed to evaluate whether CD8^+^ T cells themselves are involved in the protective mechanism seen in NTS by subjecting *CD8α^−/−^* mice to NTS with or without rIL-15 treatment (Experimental setup Figure 5A). *CD8α^−/−^* mice subjected to NTS and treated with rIL-15 showed higher albuminuria (Figure 5B) and higher PAS scores as compared to *CD8α^−/−^* mice treated with vehicle (Figure 5C,D). Of note, *CD8α^−/−^* mice treated with rIL-15 had comparable levels of circulating mouse anti-rabbit IgG (Appendix A) and comparable numbers of iNKT cells in kidney tissue (Appendix A). In addition, immunohistochemical staining revealed a trend of more Ly6G^+^ cells in glomeruli (Figure 5E) and higher numbers of interstitial Ly6G^+^ cells (Figure 5F) and no difference in numbers of CD68^+^ cells (Figure 5G) infiltrating the kidney.

*CD8α^−/−^* mice with rIL-15 treatment showed a trend of increased tubular cast formation 7 days after NTS induction (Figure 5H). TUNEL staining revealed a comparable number of TUNEL^+^ cells in kidney tissues in both groups (Figure 5I). Gene expression profiles of *CD8α^−/−^* mice revealed that in the absence of CD8^+^ T cells, neither *Il-15rα* nor *Bcl-2* is upregulated on TECs in NTS after rIL-15 administration (Figure 5J). 

To exclude influences of the knock-out strain on our results, we depleted CD8α^+^ cells by antibody in vivo after induction of NTS. WT mice were subjected to NTS and received rIL-15 treatment on day 1 after NTS induction. On day 2, one group was depleted of CD8α^+^ cells, whereas the control group received an isotype control antibody (Figure 6A). After 7 days of NTS, the CD8α-depleted group showed a trend of higher albuminuria (Figure 6B) and a trend to more glomerular scarring (Figure 6C). Tubular injury analyzed by cast formation was increased without CD8^+^ T cells present (Figure 6D).

Additionally, we conducted the adoptive transfer of CD8^+^ T cells to *CD8α^−/−^* mice subjected to NTS and treated with rIL-15. As controls, *CD8α^−/−^* mice without transfer of cells were subjected to NTS and treated with rIL-15 (Figure 6E). Transferred CD8α^+^ cells were isolated from lymph nodes of healthy WT mice and enriched to 97.2% purity (Figure 6F). Albuminuria was similar in both groups on day 7 of NTS. However, we observed a 50% reduction of albuminuria (Figure 6G) and less glomerular (Figure 6H) and tubular injury (Figure 5I) on day 14 in the group transferred with CD8α^+^ cells.

## 4. Discussion

In this manuscript, we add to existing evidence that IL-15 plays an important protective role in a murine model of immune complex-mediated GN in part by protecting tubular epithelial cells. Although further translational studies are needed, we suggest low-dose rIL-15 as a new potential treatment for GN. Additionally, we show that the protective effect of low-dose rIL-15 treatment in NTS is dependent on CD8^+^ T cells in vivo.

We found significant amelioration of NTS disease outcomes with rIL-15 treatment, which was not explained by changes in IgG deposition on the glomerular basement membrane or T cell infiltration into the kidney. However, our results show a IL-15-mediated improvement of TEC survival in NTS, as has been shown before [17]. Previous in vitro studies report an increase in survival of TECs upon stimulation with rIL-15 [16] and increased cell death in *IL-15^−/−^* TECs [17]. The same study induced NTS in *IL-15^−/−^* mice and found more tubular injury, accompanied by more glomerular damage, compared to wild-type mice. In line with these results, we now report that therapeutical treatment of wild-type NTS mice with rIL-15 reduces tubular cast formation, decreases TEC death, and improves glomerulosclerosis. There seems to be a clear association between TEC damage and glomerular injury in NTS due to the recruitment of monocytes and macrophages. Recruitment is mediated by damaged TECs which release various chemokines, such as Mcp-1 or Cxcl-1 and -5, thereby further increasing renal damage not only in the interstitium but also in glomeruli [17,36,37]. A recent study in a murine model of AKI-to-CKD transition highlighted the impact of macrophages on TEC repair [38]. Macrophage persistence after injury promoted the recruitment of proinflammatory T cells and neutrophils to the tissue and hindered repair pathways, resulting in TEC atrophy. In our hands, we see a profound decrease in the infiltration of macrophages and neutrophil granulocytes in NTS mice treated with low-dose rIL-15 without observing differences in CD4^+^ T cell numbers. Nevertheless, we did not see *Mcp1*, *Cxcl1, Cxcl5,* and *Nlpr3* mRNA changes in TEC isolates in NTS mice treated with rIL-15 compared to vehicle. Thus, further research is needed to unravel how rIL-15 inhibits the recruitment of macrophages and neutrophils and whether this effect is mediated by its impact on TECs. Additionally, we cannot exclude a direct effect of rIL-15 on macrophages and neutrophil granulocytes [39]. 

Cytokine-based immunotherapies such as low-dose IL-2 prove potent in murine models of autoimmune disease and also human diseases such as systemic lupus erythematosus (SLE) [40] and GvHD [41,42]. IL-2 and IL-15 are members of the same cytokine family, and their heterotrimeric receptors share two receptor subunits [43]. Still, their functions and target cells differ significantly. Low-dose IL-2 therapy stimulates Treg proliferation and has been successfully used in models of autoimmunity [44] and human diseases. In contrast, IL-15 targets primarily CD8^+^ T cells and NK cells, and thus high doses are used for cancer therapy [43]. The role of CD8^+^ T cells in GN is still disputed [27]. Studies conducted in autoimmune GN rodent models suggest a detrimental role for CD8^+^ T cells in disease onset and progression [45,46]. However, studies in non-autoimmune GN rodent models, like our model of NTS, report worse outcomes if CD8^+^ T cells are depleted or genetically knocked out [22,47,48]. Recently, it has been suggested in vitro that TECs also have the possibility to cross-present antigens to CD8^+^ T cells, which underlines their close interaction [40].

In NTS mice treated with rIL-15, we detected fewer IFNγ-producing CD8^+^ T cells in the spleen. When isolating pan-CD8^+^ T cells from the lymph node of these mice, we found them to express decreased mRNA transcripts of *perforin*, *granzyme B*, and *Ifnγ*. On the other hand, the transcription factor Helios (*Ikzf2*)–described as a transcription factor for regulatory functions in CD8^+^ T cells [49,50]-was increased. Interestingly, a recent study could provide evidence that patients with SLE present with fewer CD8^+^Helios^+^ regulatory T cells [51], further pointing towards the importance of this cell population in autoimmunity. IL-15 is known to promote CD8^+^ memory T cell survival [33]. IL-15 is also associated with the expression of immune checkpoints on CD8^+^ T cells, such as the production of IL-10, the expression of PD-1, and the maintenance of CD122^+^CD8^+^ regulatory T cells [52,53]. To target the later effects, we aimed to use low-dose rIL-15, by which we were able to significantly increase our CD8^+^ memory T cell population in the lymph nodes. We report an IL-15-driven expansion of CD122^+^CD8^+^ memory T cells and an upregulation of the marker Ly49. Both populations have previously been associated with presumable regulatory functions in CD8^+^ T cells [52,54,55]. Additionally, Ly49 knockout mice have been reported to develop GN spontaneously [50]. Still, the existence of this cell entity is under debate since the exact characterization of these cells is difficult. From our data, we can only extrapolate that rIL-15 treatment influenced CD8^+^ T cells towards a less cytotoxic phenotype and improved the NTS phenotype. 

In our hands, rIL-15-mediated protective effects on TECs and glomerular injury in NTS were dependent on CD8^+^ cells since IL-15 treatment was not beneficial in *CD8α^−/−^* mice with NTS. In stark contrast to our WT data, *CD8α^−/−^* TECs of mice treated with IL-15 did not upregulate the IL-15Rα-chain and *Bcl2*, a gene associated with protection from cell death. We confirmed that the beneficial effect of IL-15 in NTS depends on CD8^+^ cells, with anti-CD8α antibody-depletion in wild-type NTS mice. This approach also excludes that the lack of benefits seen in *CD8α^−/−^* mice is not due to different background strains. These results, however, contradict in vitro studies showing a survival benefit for TECs treated with IL-15 without the presence of other cell types [17]. To further underline the importance of CD8^+^ T cells in the efficacy of rIL-15 treatment in vivo, we adoptively transferred CD8^+^ T cells to *CD8α^−/−^* mice induced NTS and treated them with low-dose rIL-15. IL-15 treatment improved the NTS phenotype in CD8^+^ T cell replenished mice but in a delayed fashion, which led us to speculate on an early beneficial role for CD8^+^ tissue-resident cells, known to be present in the wild-type kidney [56]. Taken together, these results highlight the importance of low-dose rIL-15-mediated changes of CD8^+^ T cell function to improve the NTS phenotype.

Due to the multiple targets of IL-15 among immune and resident cells, evaluating the exact pathomechanism of rIL-15 therapy is complex. NK cells consist of numerous subpopulations and are well-known targets of IL-15 therapy, which is used to treat cancer [57]. At least conventional NK cells have recently been proven to be dispensable in the pathogenesis of NTS [58], whereas iNKT cells improve NTS [59]. Therefore, we focussed on evaluating iNKT cells but did not see a difference in iNKT cell numbers in rIL-15 treated mice. Additionally, CD4^+^ T cells were analyzed since they are pivotal in the NTS pathogenesis [60] and are also targets of IL-15, stimulating their proliferation and acting as a chemoattractant [57]. Interestingly, we did not see differences in CD4^+^ T cell numbers in kidneys after NTS induction, but a signal towards increased Th1 activity in the spleen when mice were treated with IL-15, which does not explain the protective effect of low-dose rIL-15 therapy in NTS. Since CD4^+^ Tregs have also been described to be influenced by IL-15 [61] and play a crucial role in the pathogenesis of NTS [24], we analyzed their abundance and ruled out an effect on Treg numbers due to low-dose rIL-15 treatment.

Our murine findings imply a potential role for low-dose rIL-15 cytokine-based immunotherapy in GN. We highlight the role of CD8^+^ T cells, however, we cannot exclude the possibility of other immune cells being involved in the IL-15-mediated amelioration of NTS. Here, further studies are necessary to unravel additional functions of low-dose rIL-15. However, since the GN patient population is broadly treated with immunosuppressants, our findings might already be important for first considerations regarding studies with a translational focus and ultimatively clinical trials.

## Figures and Tables

**Figure 1 cells-11-03656-f001:**
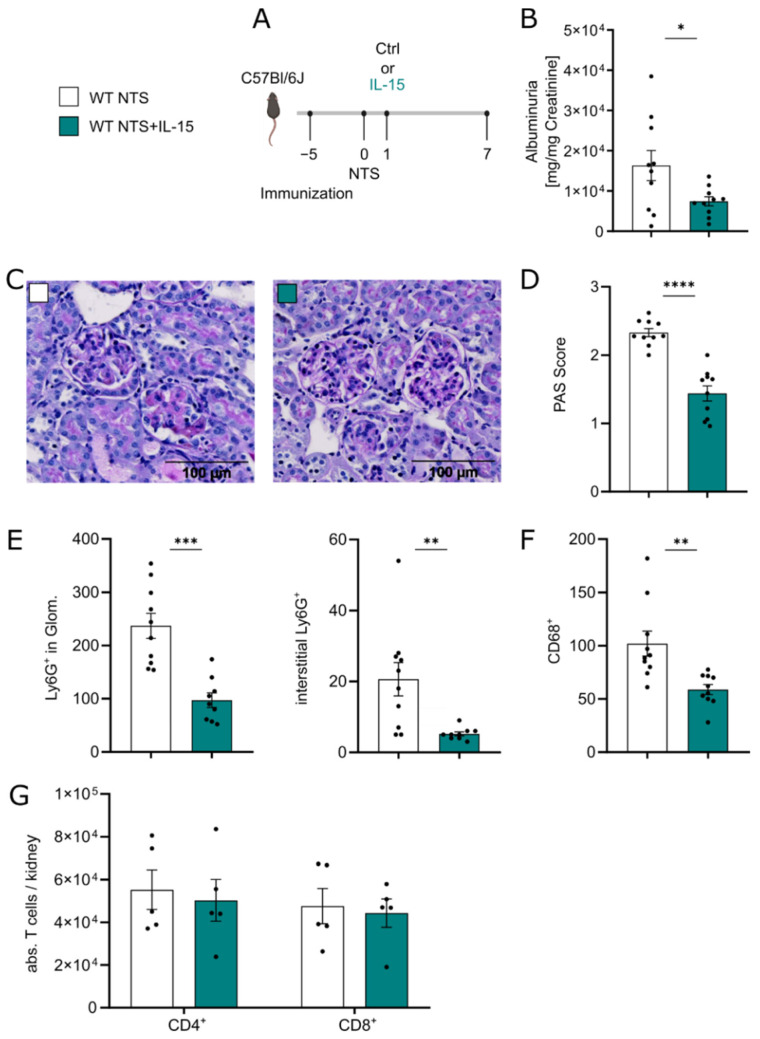
Administration of low-dose rIL-15 is protective in NTS. (**A**) Experimental timeline of 7 days NTS with single low-dose rIL-15 intervention. (**B**) Ratio of urinary albumin to creatinine. (**C**) Representative histological PAS staining (×40 magnification) and (**D**) scoring of PAS-positive material in murine kidney tissue. (**E**) Immunohistochemical quantification of glomerular and interstitial Ly6G^+^ cells and (**F**) renal CD68^+^ in 6hpf. (**G**) Flow cytometry analysis of CD4^+^ and CD8^+^ T cells in kidney tissue. Data in B-F represent two independent experiments. Black dots represent individual values of respective mice. * *p* ≤ 0.05, ** *p* ≤ 0.01, *** *p* ≤ 0.001, **** *p* ≤ 0.0001. Statistical analysis used was Student’s *t*-test or Mann–Whitney test. All data are mean ± SEM.

**Figure 2 cells-11-03656-f002:**
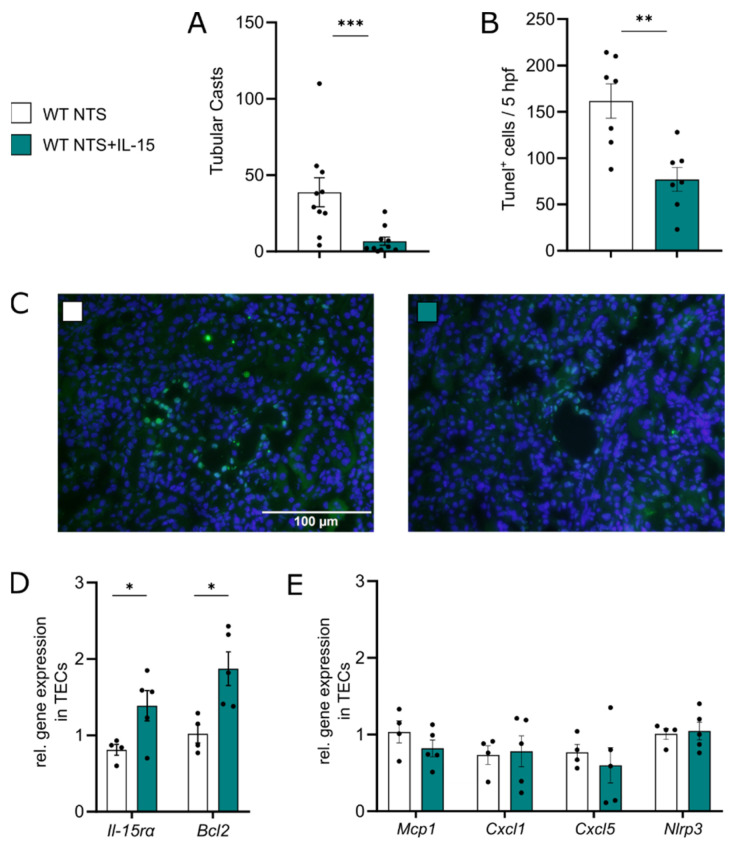
rIL-15 promotes TEC survival in NTS. Data represent day 7 of NTS. (**A**) Quantification of tubular cast formation in PAS-staining of kidney tissue. (**B**) Quantification of TUNEL^+^ cells in 5 hpfs and (**C**) representative micrographs (×40 magnification). (**D**) Relative gene expression of *Il-15rα* and *Bcl2* and (**E**) *Mcp1*, *Cxcl1*, *Cxcl5*, and *Nlrp3* in renal tubular epithelial cells. Data in A and B represent two independent experiments. Black dots represent individual values of respective mice. * *p* ≤ 0.05, ** *p* ≤ 0.01, *** *p* ≤ 0.001. Statistical analysis used was Student’s *t*-test or Mann–Whitney test. All data are mean ± SEM.

**Figure 3 cells-11-03656-f003:**
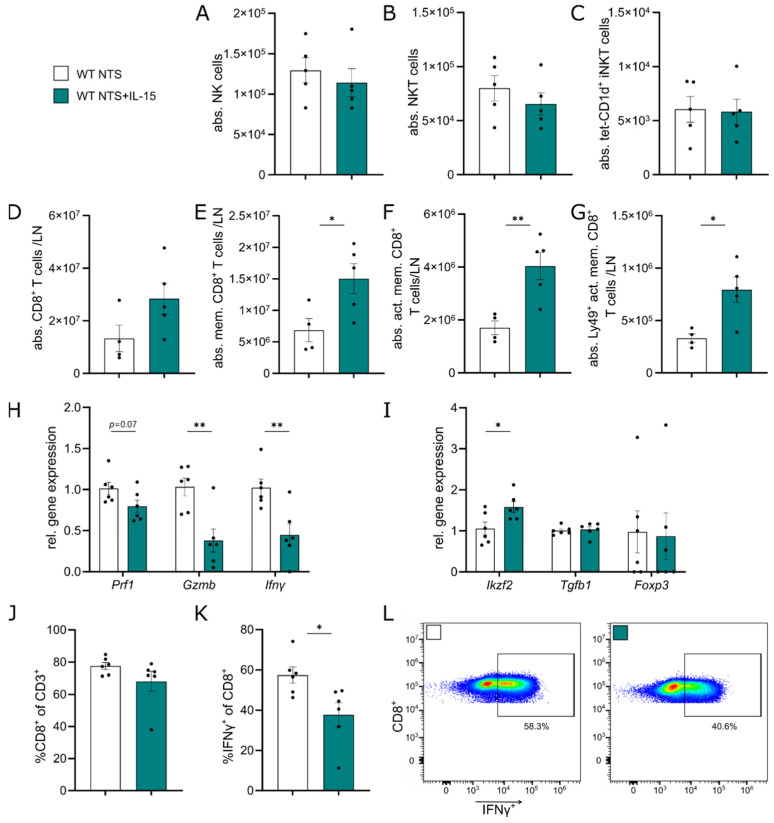
IL-15 increases CD44^+^ memory CD8^+^ T cell numbers in the periphery and mitigates a cytotoxic phenotype. Data represent day 7 of NTS. Quantification of (**A**) CD45^+^CD19^-^CD3^-^NK1.1^+^ NK cells, (**B**) CD45^+^CD19^-^CD3^+^NK1.1^+^ NKT cells, and (**C**) CD45^+^CD19^-^CD3^-^tet-CD1d^+^ iNKT cells in kidney tissue. Quantification of (**D**) CD45^+^CD3^+^CD8^+^ T cells, (**E**) CD44^+^CD8^+^ memory T cells, (**F**) CD122^+^ memory CD8^+^ T cells, and (**G**) Ly49^+^CD122^+^ memory CD8^+^ T cells in the lymph node. (**H**) Relative gene expression of *Prf1*, *Gzmb*, *Ifnγ* and (**I**) *Ikzf2*, *Tgfb1*, and *Foxp3* in sorted CD8^+^ T cells after 4 days of culture. Flow cytometry analysis of frequencies of (**J**) CD8^+^ and (**K**) IFNγ^+^ cells of CD8^+^ T cells and (**L**) representative plots of splenocytes after 4 days of culture. Data in H-L represent two independent experiments. Black dots represent individual values of respective mice. * *p* ≤ 0.05, ** *p* ≤ 0.01. Statistical analysis used was Student’s *t*-test or Mann–Whitney test. All data are mean ± SEM.

**Figure 4 cells-11-03656-f004:**
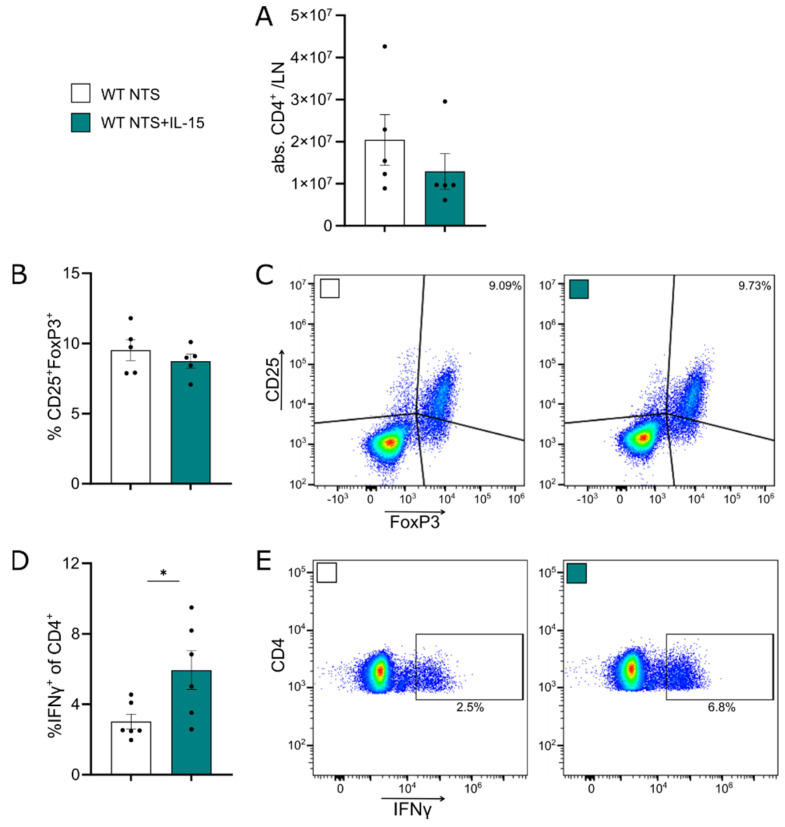
rIL-15 increases IFNγ production of CD4 T cells. Data represent day 7 of NTS. (**A**) Quantification of CD4^+^ T cells and (**B**) frequencies and (**C**) representative plots of CD25^+^Foxp3^+^ T cells in the lymph node. Flow cytometry analysis of frequencies of (**D**) IFNγ^+^ cells of CD4^+^ T cells and (**E**) representative plots of splenocytes after 4 days of culture. Data in D represent two independent experiments. Black dots represent individual values of respective mice. * *p* ≤ 0.05. Statistical analysis used was Student’s *t*-test or Mann–Whitney test. All data are mean ± SEM.

**Figure 5 cells-11-03656-f005:**
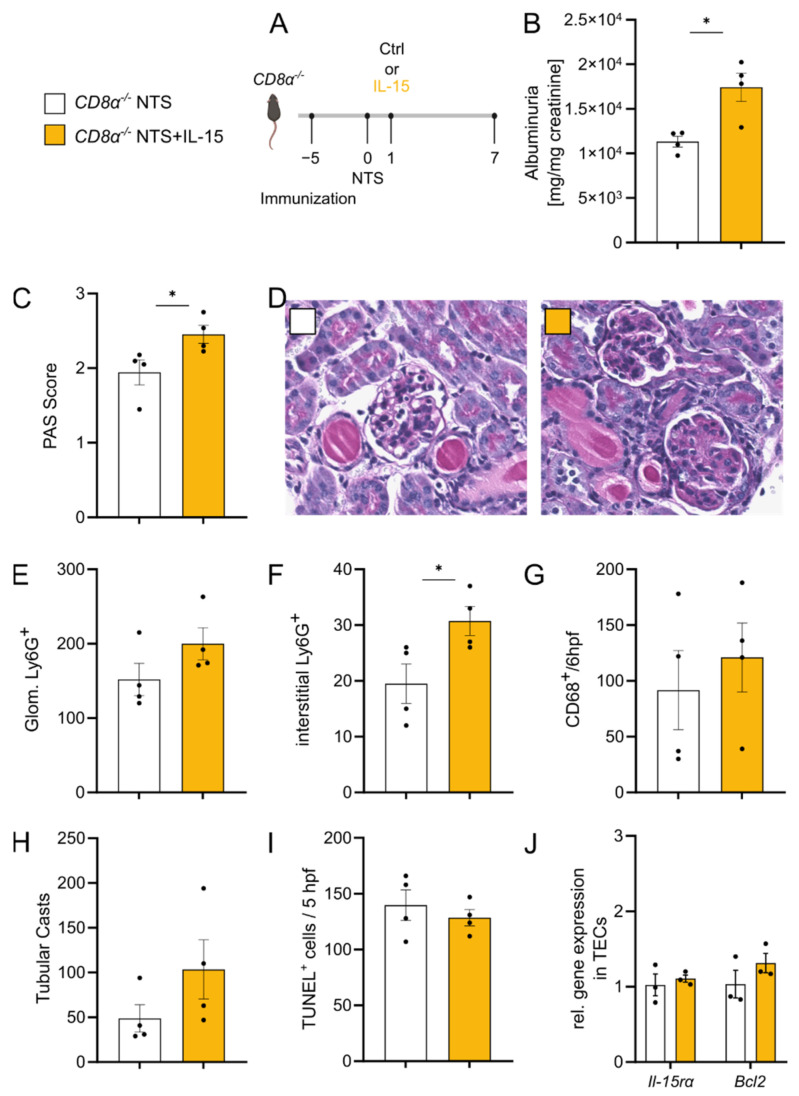
rIL-15 in NTS has no beneficial effect in *CD8α^−/−^* mice. (**A**) Experimental timeline of 7 days NTS in *CD8α^−/−^* mice with a single low-dose rIL-15 intervention. (**B**) Ratio of urinary albumin to creatinine. (**C**) Scoring of PAS-positive material and (**D**) representative staining in murine kidney tissue. (**E**) Immunohistochemical quantification of glomerular and (**F**) interstitial Ly6G^+^ cells and (**G**) renal CD68^+^ in 6hpf. (**H**) Quantification of tubular cast formation in PAS-staining of kidney tissue. (**I**) Quantification of TUNEL^+^ cells in 5 hpfs. (**J**) Relative gene expression of *Il-15rα* and *Bcl2* in renal tubular epithelial cells. Black dots represent individual values of respective mice. * *p* ≤ 0.05. Statistical analysis used was Student’s *t*-test or Mann–Whitney test. All data are mean ± SEM.

**Figure 6 cells-11-03656-f006:**
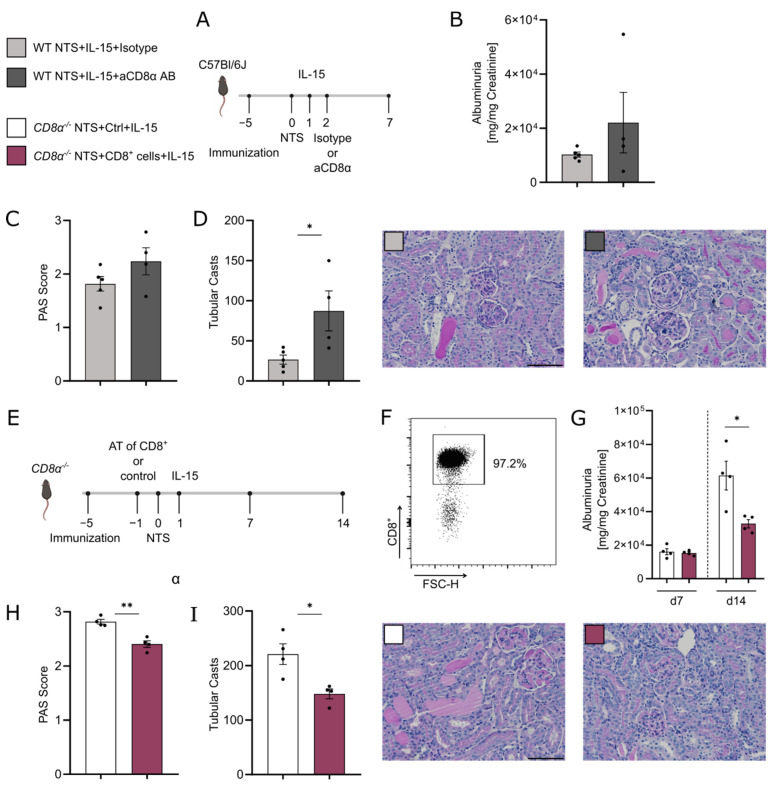
Adoptive transfer of CD8α^+^ cells into knockout mice and rIL-15 treatment reveal belated amelioration of NTS. (**A**) Timeline of the experimental setup of 7 days of NTS with low-dose rIL-15 intervention in both groups plus aCD8α antibody or isotype control antibody. (**B**) Urinary albumin to creatinine ratio, (**C**) PAS Score, and (**D**) tubular cast formation per 12 hpf in both groups and representative histological PAS stainings (×20 magnification, scale bar bottom right 100 µm). (**E**) Timeline of the experimental setup of 14 days of NTS in *CD8α^−/−^* mice. One group received 2.5 × 10^6^ CD8α^+^ cells on day –1, and both groups were treated with low-dose rIL-15 on day 1 of NTS. (**F**) Dot plot of sorted and adoptively transferred CD8α^+^ cells showing 97.2% purity of cells. (**G**) Urinary albumin to creatinine ratio on days 7 and 14 of NTS, (**H**) PAS Score, and (**I**) tubular cast formation per 12hpf on day 14 of NTS and representative histological PAS stainings (×20 magnification, scale bar bottom right 100 µm). Black dots represent individual values of respective mice. * *p* ≤ 0.05, ** *p* ≤ 0.01. Statistical analysis used was Student’s *t*-test or Mann–Whitney test. All data are mean ± SEM.

## Data Availability

Data is contained within the article or Appendix A.

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
