# Peer review of "Low-Dose rIL-15 Protects from Nephrotoxic Serum Nephritis via CD8+ T Cells"

_cells, 2022, doi:10.3390/cells11223656_

Round 1

Reviewer 1 Report

Well written manuscript that merits in my view publications. Mine are just a few commentaries with the intent to make the MS better.

I would suggest to keep in the area of discussion and in their final statements, clarity on the fact that these are mainly murine findings. And there is still a long way to go, hence, some humbleness on statements. On the translational area, it would have been great to see experiments in human renal cell cultures. 

Author Response

We thank the Reviewer for his/her positive statement about our manuscript. We absolutely agree with the Reviewer that there is still a long way to the translation into humans and we might have been too optimistic in our discussion since we only show murine GN data. We thus re-formulated our last sentence of the discussion. 

In vitro data on direct IL-15 effects on human renal epithelial cells have already been published, but we agree that there will be additional experiments needed to set the stage for a clinical trial in humans. We nevertheless believe, that there is a complex interaction between various cell populations in case of low-dose rIL-15 treatment, which is complex to study in vitro. 

Reviewer 2 Report

Mooslechner et al studied the effects of administration of low-dose rIL-15 in a nephrotoxic serum nephritis model in C57BL6/J mice. The administration of rIL-15 on day 1 of the model resulted in lower albuminuria, less macrophage infiltration and less histological damage which appeared to go hand in hand with numbers of CD8+ cells. This was then tested in a CD8-/- TEC knock-out model. The authors hypothesize these effects to be due to higher TEC survival but could not find higher rates of chemokines mediating this effect. 

To be honest, this paper seems to have undergone decent revisions already and reads very well. The experiments are clear and logical, and the authors confirm an important role for IL-15. 

I have no further comments and would support publication. 

Author Response

We really thank the Reviewer for his/her generous comment. Yes, it was a long way and we already worked a lot on this manuscript.

Reviewer 3 Report

Clarify the  terms  “the glomerular phenotype”  (line 377):   the glomerular phenotype probably corresponds to glomerulosclerosis which is evaluated by PAS SCORE

The authors demonstrated a profound decrease in the infiltration of macrophages and neutrophil granulocytes in NTS mice treated with low-dose rIL-15 without observing differences in CD4+ T cell numbers. In 2022 , Leyuan Xu et al.  demonstrated , in a model of  AKI-to-CKD transition , that macrophage persistence after injury promotes a T cell- and neutrophil-mediated proinflammatory milieu and progressive tubule damage.   Could the authors comment briefly their results in light of this knowledge?

Author Response

We thank the Reviewer for his/her positive comments on our manuscript. 

Line 377: We agree and changed to glomerulosclerosis

Line 381: We included and shortly discussed the finding of Leyuan from 2022 as suggested by the Reviewer. in contrast to Leyuan, we did not see differences in T cell infiltration profiles and our follow up is too short to really see any fibrosis. Nevertheless, we can not exclude that this mechanism might be in place in the long time follow up and contribute to the positive effects of low-dose rIL-15.